# Simulation of Soluble and Bound VEGF-stimulated in vitro Capillary-like Network Formation on Deformed Substrate

**Hsun Chiang, Chih-Ang Chung** *

Department of Mechanical Engineering, National Central University, Taoyuan, Taiwan

* cachung@ncu.edu.tw

**Data Availability Statement:** All data and code used for running simulations, model fitting, and

## Abstract

Capillary plexus cultivation is crucial in tissue engineering and regenerative medicine. Theoretical simulations have been conducted to supplement the expensive experimental works. However, the mechanisms connecting mechanical and chemical stimuli remained undefined, and the functions of the different VEGF forms in the culture environment were still unclear. In this paper, we developed a hybrid model for simulating short-term *in vitro* capillary incubations. We used the Cellular Potts model to predict individual cell migration, morphology change, and continuum mechanics to quantify biogel deformation and VEGF transport dynamics. By bridging the mechanical regulation and chemical stimulation in the model, the results showed good agreement between the predicted network topology and experiments, in which elongated cells connected, forming the network cords and round cells gathered, creating cobblestone-like aggregates. The results revealed that the capillary-like networks could develop in high integrity only when the mechanical and chemical couplings worked adequately, with the cell morphology and haptotaxis driven by the soluble and bound forms of VEGF, respectively, functioning simultaneously.

## Author summary

Microvascular incubation is crucial in regenerative medicine and tissue engineering because of its connection between engineered organizations and the living body. Static incubation in a Petri dish provides an effective way to observe the cell migration during blood vessel formation and record the cell behaviors in different environments: various coating substrates on the Petri dish surface and multiple additives in the culture medium. In this paper, we built a mathematical model based on the interactions between the endothelial cells, the biogel components, and the specific chemicals that regulate blood vessel development. Our simulation results showed good agreement with the literature experiments, both in the single cell shape and overall cell arrangement, revealing that the stable capillary network resulted from the mechanical attachment and traction between cells and substrate materials and the chemical stimulations by spatial signals.

plotting is available on Zenodo: https://zenodo.org/records/10902008.

**Funding:** H.C. was funded by the National Science and Technology Council, Taiwan under the grant (MOST 107-2221-E-008-057-MY3). C.A.C. was funded by the National Science and Technology Council, Taiwan under the grants (MOST 107-2221-E-008-057-MY3, MOST 111-2221-E-008-069-MY3). The funder had no role in the design of the study, data collection, analysis and interpretation, decision to publish, or preparation of the manuscript.

**Competing interests:** The authors have declared that no competing interests exist.

## 1. Introduction

Capillary development, such as in angiogenesis and vasculogenesis, plays a crucial role in embryonic development and wound healing, as well as the realization of regenerative medicine and tissue engineering. Cultivating blood capillaries is vital for the circulation connection and waste removal between artificial tissue and the living body [1–3]. Capillary cultivation *in vitro* can be classified into two categories depending on the experimental settings and lasting times [4]. The short-term cultivations typically last from a few hours to days. The necessary chemicals during the capillary developing process are offered well before the trial startup. Such tests usually aim to research the endothelial cell migration behavior or the interactions between the cells and their environments [5–7]. The long-term incubations, on the other hand, are generally over weeks or months to allow for the extracellular matrix (ECM) or the functional proteins secreted from cells, mimicking the realistic environments for capillary growth and showing potential for applications such as wound healing and transplanting [8–10].

Since the 1980s, researchers have tried various substrates in *in vitro* capillary incubating experiments. They studied the relationships between endothelial cell behaviors and the mechanical properties of biogel materials. These studies revealed biogel remodeling, in which the cells exert traction forces on the substrates and disturb the material density during cell migration [5,9,11]. For different purposes, researchers have designed various biogels composed of specific components, such as the mixing of collagen and gelatin to simulate the environments where proteins are denatured in burns and scalds [12] or the coated fibronectin on scaffolds to study the fibronectin structures and functions for cell attachment promotion [13,14]. Fibronectin is a biomaterial widely studied in recent years for its integrin-binding sub-domains, which can bind the integrin on the cellular membrane to stabilize the cell attachment. Moreover, together with its growth factor-binding sub-domains corresponding to various growth factors, fibronectin regulates growth factor sources in both *in vitro* and *in vivo* experiments [15–18].

Later on, thanks to the development of protein purification techniques, the mainstream research topics shifted to the endothelial cell responses induced by vascular endothelial growth factors (VEGFs). The VEGF research can be traced back to 1977 when the vascular permeability factor (VPF) was discovered around the tumor cells to increase the capillary permeability and obtain extra nutrients [19]. Studies have confirmed that VEGFs present multiple functions for regulating endothelial cell growth, proliferation, migration, and morphology. VEGFs comprise five isoforms: the A-, B-, C-, and D-type and the placental growth factor (PGF), which induce the differentiation of endothelial cells into blood or lymphatic vessels [20,21]. Among the five isoforms, the VEGF-A, comprising the most abundant 165-amino acids in the human body, is most related to capillary development [22]. Furthermore, the VEGF binding to substrates is commonly called the bound VEGF, which is considered to dominate the angiogenesis and vasculogenesis [23–25]. Besides, experimental results in recent years showed that the synergy between integrin and bound VEGF on fibronectin enhances the VEGF stimulation on endothelial cells, further promoting angiogenesis [26].

In addition to experiments, theoretical models and numerical simulations have been conducted to study capillary development. The simulation models have two categories: continuum- and cell-based models, according to their calculation methods for cell behaviors. The continuum-based model uses partial differential equations (PDEs) to describe the cell behaviors in a continuous spatial distribution. Murray and Oster first established an equation set that includes the dynamics of cell colony and extracellular matrix (ECM). Considering the force balance between the cells and ECM, they performed the linear stability analysis to explore the cellular network conditions [27]. Manoussaki *et al.* followed and simplified Murray and

Oster's equations into two-dimensional by considering the migration of endothelial cells limited on the upper layer of the matrix, successfully obtaining the mechanical force-induced visualized pattern of non-uniform distributed cells [28]. Namy *et al.* added the long-ranged elasticity descriptions for the matrix fibers to the theoretical model, making the calculation results more quantitatively comparable between experiments and numerical simulations [29]. In addition, Gamba *et al.*, in the era of growing understanding of VEGF, assumed VEGF to be the driver of cell migration and obtained the chemical signal-induced cellular networks [30].

Cell-based models, such as the cellular Potts model (CPM), were developed to consider cell individual behaviors using a probability approach rather than that of the cell colony. The CPM framework, viewing individual cells moving on a fixed grid, can be traced back to Graner and Glazier, who extended the Potts model with several constraints suitable for living cells, and the cell alliances tended to be stable [31,32]. Then, Merks *et al.* coupled both the cell-based and the continuum-based models. In their hybrid approach, CPM predicted the endothelial cell migration, while PDEs calculated the VEGF concentration [33]. Scianna and Munaron further included the intracellular calcium signaling induced by VEGF in their model, linking the VEGF to the cytoskeletal remodeling and elongation phenomena of endothelial cells [34]. Later, the VEGF sources were modified from the autocrine assumption to paracrine based on experimental evidence. Köhn-Luque *et al.* assumed that haptotaxis induced by the bound form of VEGF dominated the cell migration rather than the chemotaxis generated by the soluble form of VEGF [35]. Lima *et al.* combined the macro tumor with the micro sprouting models, suggesting that VEGF para-secreted by tumor cells provided endothelial proliferation [36].

On the other hand, van Oers *et al.* published a mechanical-based hybrid model with their successful simulations of cell networks and sprouts, showing that the capillary-like structures could develop via the pure stress and strain mechanisms between cells and substrates [37]. After considering several phenomena, such as vascular adaptation, vessel remodeling, and capillary collapsing, Vilanova *et al.* developed a model based on cell movement, expansion, and apoptosis, simulating the capillary growth consistent with the experimental data [38]. In recent years, Phillips *et al.* combined the confocal experimental data with computational simulations, predicting the total sprouting length via calibrating and verifying endothelial cell growth cycles [39,40].

The complete driving mechanisms underlying the capillary development remain unclear. There was a consensus that the bound VEGF located on the substrates dominates the developing; however, the role of the soluble VEGF in the culture medium was commonly ignored [23–25,35]. Most mathematical models included chemical or mechanical stimulation in the development separately; the integrated connection between the mechanical and chemical mechanisms was still absent [29,41,42]. Besides, although the computational results met experiments in certain conditions, several problems remain. For instance, mechanical mechanisms omitted the necessity of bound VEGF [42], whereas chemical mechanisms could not fully reproduce the results of the tilted biogel experiments [43].

This paper studies how mechanical regulation and chemical stimulation simultaneously work in capillary network development. We developed a hybrid model using cell-based CPM to predict the endothelial cell migration and morphology and using continuum-based PDEs to quantify the biogel deformation and VEGF transport dynamics. The simulation settings and the capillary-like network results were referred mainly to the short-term experiments in the literature [44] without considering endothelial cells secreting and intaking VEGF. Close attention was paid to the impact of VEGFs in the solute and bound form, respectively. In the following, we first simulate an experimental case to ensure the correction of the computation in **Section 2**. We show how endothelial cell elongation induced by soluble VEGF and haptotaxis driven by bound VEGF regulate the network-forming process. We then performed

parameter analysis for the network formation under various mechanical and chemical mechanisms. Discussions on the model features and limitations are in **Section 3**, and the summarized conclusions are in **Section 4**. Finally, we specify the mathematical formulation and computational procedure in **Section 5**.

## 2. Results

### 2.1. Cellular network formation

Previous experimental studies showed capillary-like networks form at 18 hours of incubation [44]. **Fig 1** compares the networks of the reference result in the literature and our present simulation. **Table 1** presents the simulation parameters for their typical values and citations. The overall network topology agreed well. The local cell alliances were similar, too, exhibiting two regional cell colony features. First, the elongated cells connected head-to-tail and lined up, creating the network cords. Second, the remaining flattened, round cells gathered closely, forming a cobblestone-like aggregate. [7,45,46].

**Fig 2** shows the computed cell distribution, the biogel density, and the bound VEGF concentration through the network formation process. Because of the diffusion effect, the simulated soluble VEGF was virtually uniform across the Petri dish. We thus omit to show the soluble VEGF results. The cell network evolution and features at different times were consistent with the experimental data [44]. Initially, the cells spread randomly, and the biogel and soluble VEGF were uniform in space. The cells migrated and formed into the cellular cord elements after the first 6 hours. Then, the cellular cord elements were connected, and the cords enclosed the cell-free lacunae through 6 to 24 hours of incubation. The cell patterns altered less after 18 hours, which indicates a stable network had developed. The overall cellular network was fully developed with 1 to 3 cells in cord width, as shown in **Figs 1B** and **2A**, which agreed with the experimental observations and could meet the sequential cord hollowing requirement in angiogenesis and vasculogenesis [44,52].

The results present a strong correlation between the cells, the biogel, and the VEGF fields, which formed a positive feedback loop among the cells, biogel, and bound VEGF. Once the endothelial cells were attached to the biogel surface, the gel, as pulled by the cell traction, started concentrating, as observed during the first few Monte-Carlo steps in the simulation. In the next stage, the concentrated biogel led to the concentrated bound VEGF; the higher the biogel density, the more the VEGF binding sits. Due to haptotaxis driven by bound VEGF, the cells moved toward the more bound VEGF areas, leading to further concentrated biogel. As

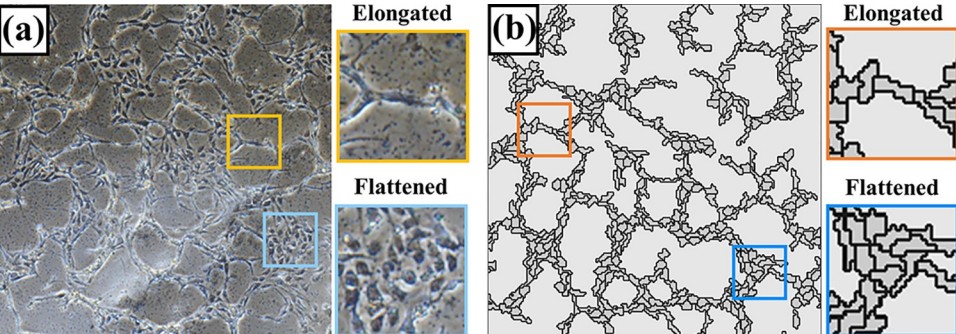

**Fig 1. Comparison between the literature experimental and current simulative cellular networks.** (a) The experimental result was obtained at 18 hours of incubation of HUVECs [44]. (b) The simulative cell distribution at 540 MCS (18 hours). Both diagrams show the cords connected by elongated cells and the cobblestone-like alliances composed of flattened round cells.

**Table 1. Parameter values.**

| Symbol | Description | Estimation | Reference |
|---|---|---|---|
| $J_{11}$ | Interface energy (cell-cell) | 0.12 a. e. u./μm | Merks et al. [41] |
| $J_{10}$ | Interface energy (cell-empty) | 0.06 a. e. u./μm | Merks et al. [41] |
| $J_{01}$ | Interface energy (empty-cell) | 0.06 a. e. u./μm | Merks et al. [41] |
| $J_{00}$ | Interface energy (empty-empty) | 0 | Merks et al. [41] |
| $\lambda_a$ | Restraint for cell area | $1.0 \times 10^{-4}$ a. e. u./μm$^4$ | This work |
| $\lambda_p$ | Restraint for cell perimeter | $1.0 \times 10^{-2}$ a. e. u./μm | This work |
| $\lambda_l$ | Restraint for cell length | $5.0 \times 10^3$ a. e. u. μm | This work |
| $\mu_m$ | Morphology strength | $1.3 \times 10^8$ μm$^3$/ng | This work |
| $\mu_h$ | Haptotaxis strength | $1.5 \times 10^{14}$ a. e. u. μm$^3$/ng | This work |
| $\kappa$ | Cell traction force strength | $4.0 \times 10^9$ Pa·μm$^3$/ng | This work |
| $E$ | Biogel Young's modulus | $2.0 \times 10^4$ Pa | Lopez-Garcia et al. [47] |
| $\upsilon$ | Biogel Poisson ratio | 0.48 | Castro et al. [48] |
| $\mu_1$ | Biogel shear viscosity | $7.4 \times 10^5$ Pa·sec | Knapp et al. [49] |
| $\mu_2$ | Biogel bulk viscosity | $1.0 \times 10^7$ Pa·sec | Gudapati et al. [50] |
| $\beta_1$ | 1$^{st}$ long-range elastic coefficient | $1.0 \times 10^4$ μm$^2$ | Tranqui and Tracqui [51] |
| $\beta_2$ | 2$^{nd}$ long-range elastic coefficient | $1.0 \times 10^4$ μm$^2$ | Tranqui and Tracqui [51] |
| $D_c$ | Diffusion coefficient for soluble VEGF | $5.9 \times 10^1$ μm$^2$/sec | Köhn-Luque et al. [42] |
| $k_{on}$ | Binding rate between VEGF and biogel | $8.6 \times 10^5$ μm$^3$/ng·sec | Köhn-Luque et al. [42] |
| $k_{off}$ | Unbinding rate between VEGF and biogel | $3.6 \times 10^{-3}$ sec$^{-1}$ | Köhn-Luque et al. [42] |
| $\gamma$ | VEGF binding site ratio in biogel | $5.0 \times 10^{-6}$ | This work |
| $\rho_0$ | Biogel initial density | $1.0 \times 10^{-5}$ ng/μm$^3$ | Chiang et al. [44] |
| $h_0$ | Biogel initial thickness | $3.0 \times 10^2$ μm | Chiang et al. [44] |
| $c_0$ | Soluble VEGF initial concentration | $2.0 \times 10^{-11}$ ng/μm$^3$ | Chiang et al. [44] |

the process continued, the elevation of biogel density and bound VEGF concentration would finally reach their quasi-equilibrium levels owing to the material deformation limitations and the force equilibrium, respectively, and the quasi-steady co-localization of cells, biogel, and bound VEGF appeared.

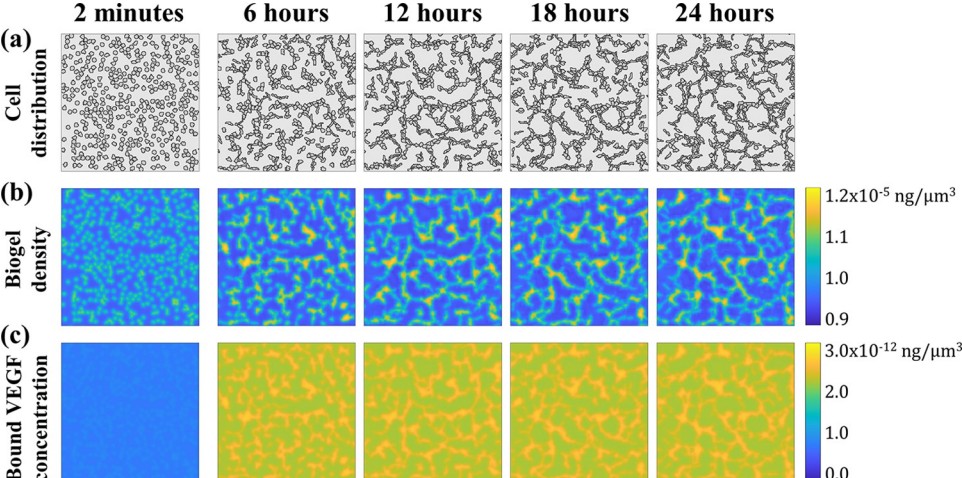

**Fig 2. Evolution of the cell distribution, biogel density, and bound VEGF concentration with incubation time.** The initial distributions were shown as 2 minutes. (a) The cell cords formed at 6 hours, connected sustainably during 6 to 18 hours and stabilized between 18 and 24 hours. (b) The biogel was locally concentrated by cell traction. (c) Bound VEGF on the biogel was concentrated following the biogel deformation.

## 2.2. Effects of soluble and bound VEGF

To investigate the roles of the two VEGF forms in the network developing process, we compare different network features by numerically turning on/off the cellular morphological change and haptotaxis effects. **Fig 3** shows the cell distributions, in which significant differences appear among the four simulation conditions. Benefiting from the periodic boundary settings for simulation, we could periodically arrange four identical computed cell images to become a 4-times figure, which could more effectively show the complete network lacunae surrounded by cellular cords than the original single figure.

Unlike the typical case with both the cellular morphology and haptotaxis turned on (**Fig 3A**), the cellular cords composed of flattened, round cells could hardly connect well to develop into a complete network pattern if we turned off the cell morphology effect induced by soluble VEGF as shown by the two cases with the cellular morphology-off (**Fig 3B and 3D**). In the two cases with haptotaxis-off (**Fig 3C and 3D**), cell migration would only primarily follow random walks because of no spatial cues from the bound VEGF. Because the elongated cells could more easily touch each other than the round cells, a randomly constructed network formed in the case with morphology turned on (**Fig 3C**). If we further turned off the morphology effect, the cells were virtually in a spread distribution (**Fig 3D**).

For comparison, the numbers of network junctions and segments and the total segment length from the reference experiment [44] against incubation time are shown in **Fig 4A**, and those from the current simulation images are shown in **Fig 4B**. The cellular network data evaluated using Image J also showed an agreement between the simulations and the previous experiments. The more complete the network was, the larger the three indicators were. The results revealed the necessity of the soluble and bound VEGF in the network-developing process. Capillary-like networks grew continuously and maintained stability only when both the VEGF forms worked appropriately. If we, in simulation, canceled the cell elongated effects induced by the soluble VEGF, the cell cords developed due to aggregation in the early stage were hardly connected. Accordingly, all three indicators were at low levels across the entire period. If we only canceled the cell gathering effects signaled by the bound VEGF, the network

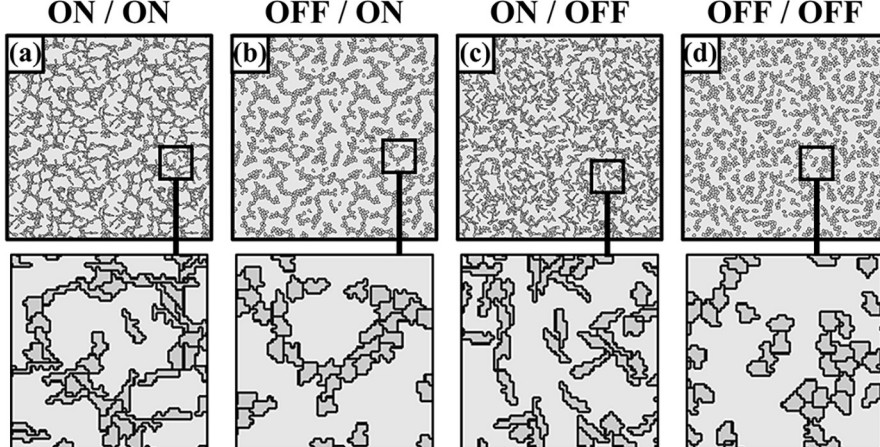

**Fig 3. Cell distribution pattern under morphology and haptotaxis effects turned on/off.** The results show the complete network formed when morphology and haptotaxis effects were both on (case a). The un-connected cell cords occurred because of insufficient cord length in the two morphology-off cases (b and d). The randomly unsustainably connected cords happened in the two haptotaxis-off cases (c and d).

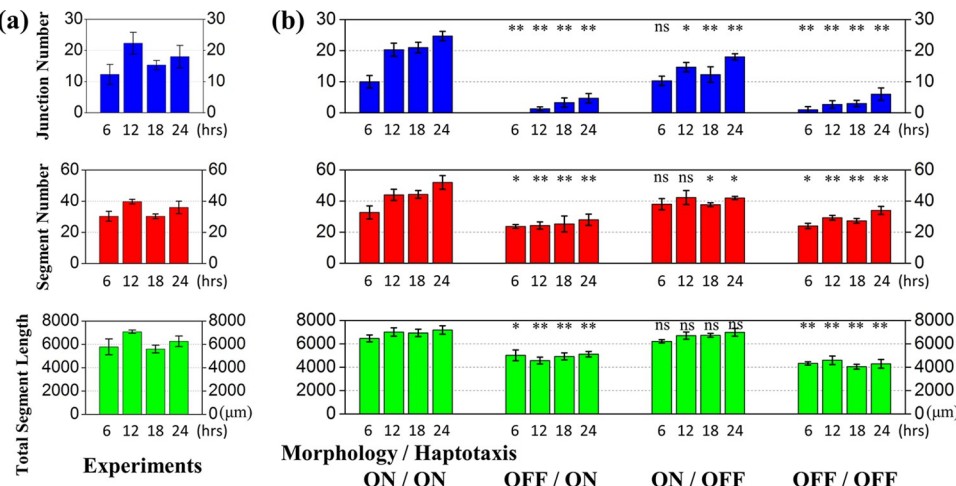

**Fig 4. Junction number, segment number, and total segment length of the capillary-like network against incubation time.** (a) Data from literature experiments [44]. (b) Data from the current simulations under morphology and haptotaxis effects turned on/off. The simulations show networks were well-developed only when the morphology and haptotaxis mechanisms worked appropriately. The junction number, segment number, and total segment length in the two morphology-off cases were significantly lower than the two morphology-on cases. The junction number in the morphology-on/haptotaxis-off was considerably lower than in the morphology-on/haptotaxis-on. Each simulation case was repeated three times, and the data are presented as mean with standard deviation. The Student's t-test was performed towards the morphology-on and haptotaxis-on case: * = p < 0.05, ** = p < 0.01, and ns = non-significant.

would show a significantly lower value in junction number than in the segment number and total segment length because the coincident contact between the elongated cells would generate segments.

We also analyzed the simulated cell shapes. As shown in **Fig 5**, we first evaluated the roundness index and aspect ratio for 400 cells every six incubation hours under morphology-on and haptotaxis-on. The results showed the increasing features of the elongated cells. The number of cells with a roundness index less than 0.3 grew significantly over time, as well as an aspect ratio greater than 3. **Fig 6** shows the roundness and aspect ratio data collected at 24 hours for various morphology/haptotaxis turned on/off. The cells remained round if the morphology effect was absent; almost no cell presented a roundness index of less than 0.3, and more than 75% of cells had an aspect ratio between 1 and 2. No significant difference in cell shape occurred between the haptotaxis on and off cases, indicating that the morphology change cue from soluble VEGF was the primary cause for cell shape distribution.

## 2.3. Modality of cell migration

The discrete CPM could trace the movement of each cell. The cell volume center was recorded in each MCS to indicate its moving direction, and its orientation through time was grouped into 12 direction intervals for statistical analysis (**Fig 7A**). We recorded the accumulated steps a cell moved in certain direction intervals. The dip test was conducted to determine the moving direction modality for a cell during numerical cultivation. The uniform and the unimodal distributions were tested sequentially using $\alpha_1 = \alpha_2 = 0.1$ [53]. **Fig 7B** shows the three typical migration modalities, where the isotropic mode refers to the cell performing pure random walks with no preferred directions, the one-direction mode represents the cell likely to migrate toward a specific direction significantly in addition to random walks, and the head-and-tail means the cell migrating mainly along the two opposite directions. The results agreed with the experiment observation for the elongated cells [7,54,55].

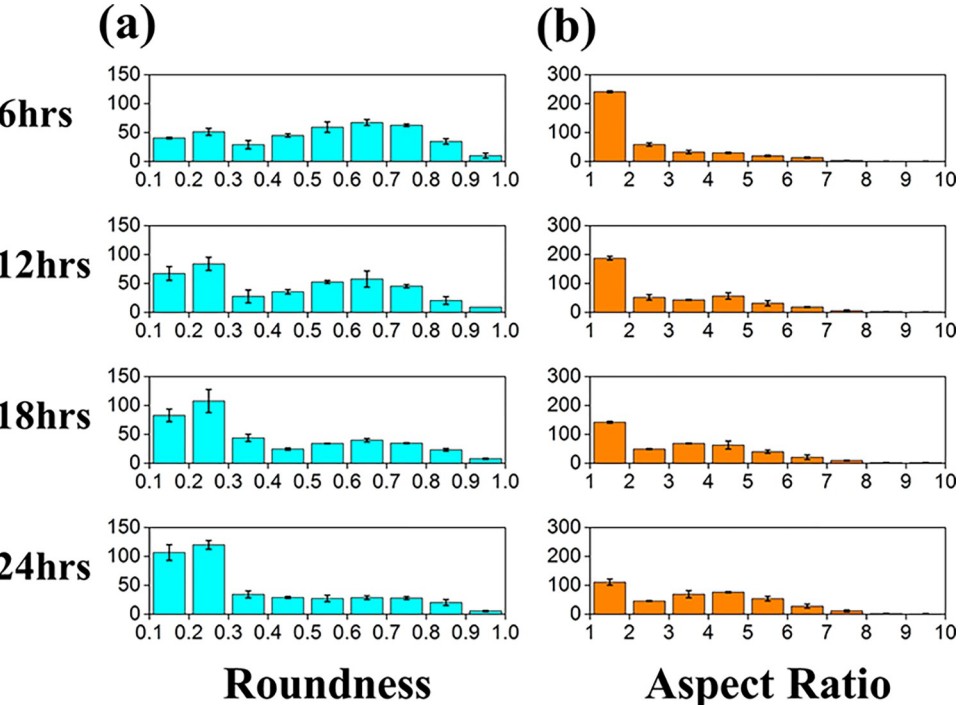

**Fig 5. Histogram of cell number versus (a) cell roundness and (b) cell aspect ratio at every 6 hours of incubation in the morphology-on/haptotaxis-on case.** The cell roundness index, unity for an exact round shape, decreases when the cell elongates. The cell aspect ratio, unity for a round shape, increases when the cell elongates. The results show that the number of cells increased with time as they lost roundness, and the number of cells that kept aspect ratio of unity decreased with time.

We counted the number of cells in different migration modalities on a two-hour base. The classification among 400 cells in each time slot is shown in **Fig 8**. For the two cases having the haptotaxis effect, the number of cells moving in one-direction mode was slightly more significant during the first few hours due to the cell aggregation from separated to the cord structures. For the two cases allowing cell morphology change, the cells appeared in the head-and-tail trend at 6 hours until the cells got elongated. It was worth mentioning that about 350 cells were showing no directional favors because of the intense random walks in the first few hours and the stable network development in the end. Approximately 40 cells appeared in the one-direction mode, and the cells in the head-and-tail mode were the fewest.

## 2.4. VEGF parameter analysis

Five parameters related to the VEGFs in the simulation are the cell morphology change coefficient $\mu_m$, the haptotaxis coefficient $\mu_h$, the initial solute concentration $c_0$, the VEGF binding rate $k_{on}$ and unbinding rate $k_{off}$. The first two parameters regulate the strength of the chemical cues, and the other three balance the two forms of VEGF contents. The total number of junctions was used to present the cellular network complexity. The junction number is better than the segment number and length in assessing the network integrity, as shown in **Fig 4,** because the elongated cells might isolated in the cavities, which should not contribute to the network integrity; however, they would be counted in the segment number and length. **Fig 9** shows how the junction number at 24 hours changed for varying one of the five parameters. We conducted regression fits based on the observed data trends. The junction number at 24 hours approached an asymptotic value with increasing parameters. Thus, the regression curves take

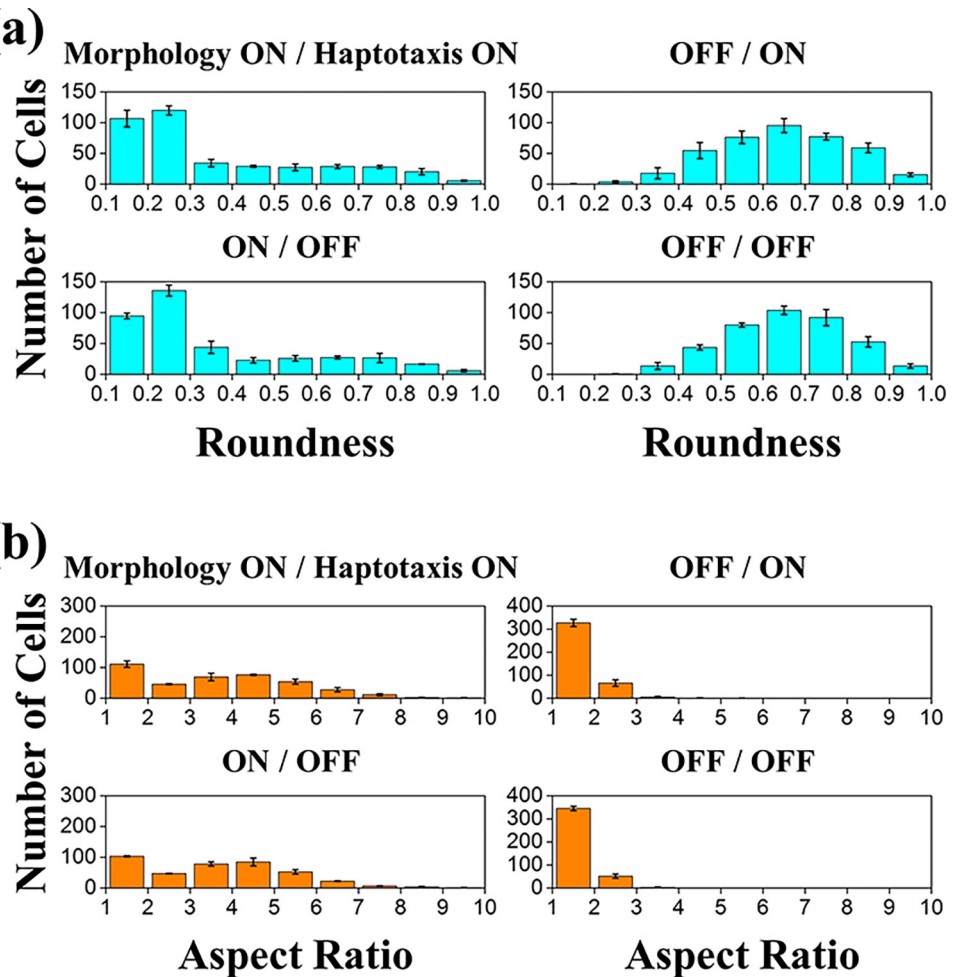

**Fig 6. Histogram of cell number versus (a) cell roundness and (b) cell aspect ratio at 24 hours under morphology and haptotaxis effects turned on/off.** Comparisons show more cells remained close to a round shape in the two morphology-off cases. In contrast, the haptotaxis turned on/off had little effect on the cell morphology change.

the exponential form as follows:

$$f(P) = a - be^{-cP} \tag{1}$$

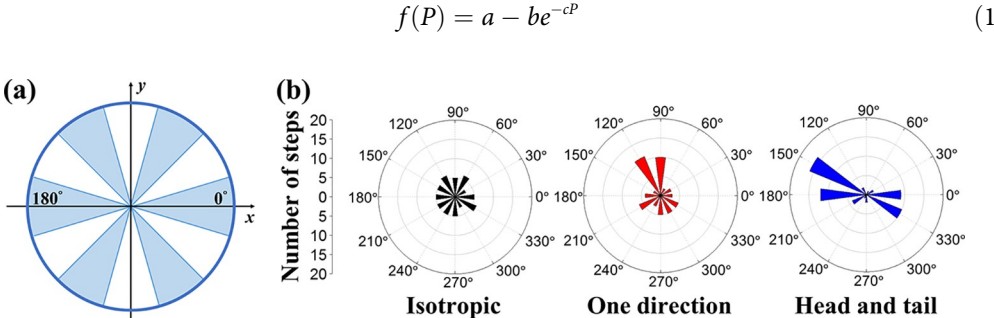

**Fig 7. Cell migration modality in the morphology-on/haptotaxis-on case.** (a) The 12 cell-center movement orientation intervals for each migration step. (b) A cell could statistically manifest one of the three migration types: isotropic, one-directional, and head-and-tail movement over 60 migration steps within two hours.

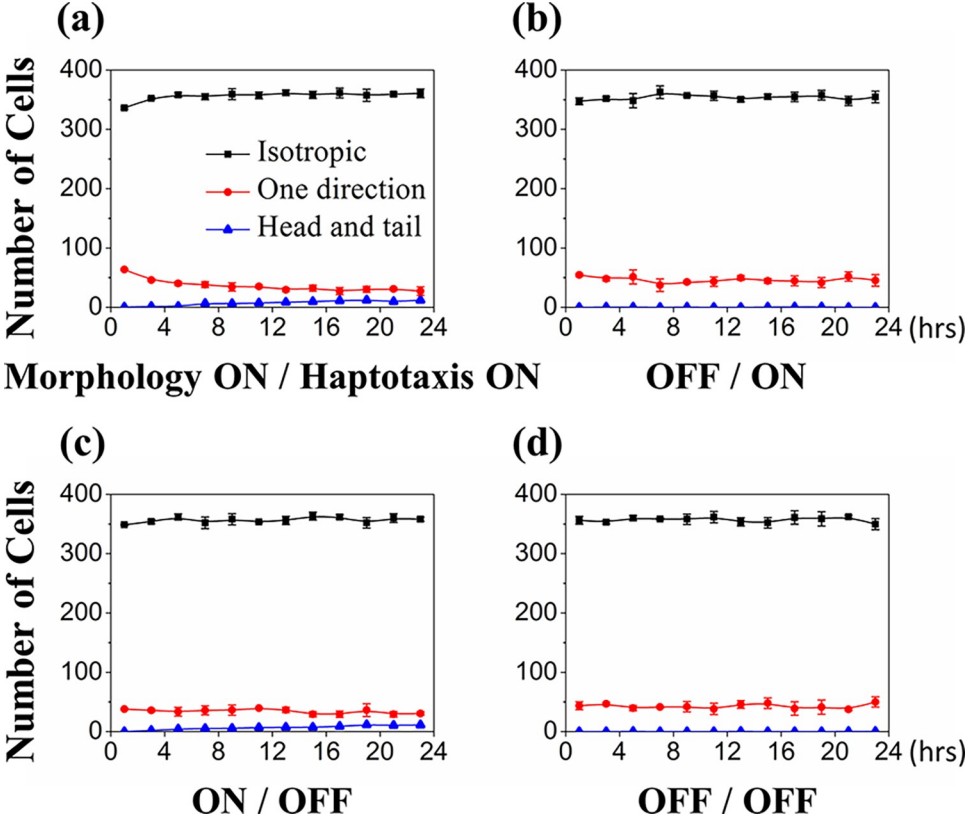

**Fig 8. Cell migration modality versus time under morphology and haptotaxis effects turned on/off.** The classification was conducted on a two-hour basis among 400 cells. The results show that most cells performed random walks in the isotropic movement type for the four simulation cases. In the two haptotaxis-on cases (a and b), the one-directional movement cells (red) were slightly more active in the first few hours. In the two morphology-on cases (a and c), the head-and-tail movement cells (blue) appear in the late hours. Each simulation was repeated three times for the same parameters.

where $P$ represents the five parameters, and $a$, $b$, and $c$ are the corresponding fitting coefficients whose values are listed in **Table 2**.

The networks at low levels of the cell morphology coefficient $\mu_m$ in **Fig 9A** presented small junction numbers, showing the necessity of the cellular elongated effects in the network formation because the elongated cells had more opportunities for the cord connection and contributed to the junction contexture. The junction number reached an asymptotic plateau value when $\mu_m$ was sufficiently great, indicating the limiting network complexity under the given cell number and the strength of the other parameters. Other curves in **Fig 9** exhibited similar trends, revealing the network integrity depended on the haptotaxis strength and the sufficient amount of soluble and bound VEGF to support the network development.

Besides the junction number trends, the simulated cellular network patterns are shown in **Fig 9** for 2x and 4x the typical parameter values. The number of incomplete network loops decreased as the junction number increased, and there was almost no open loop if the total junction number was higher than 170, which appeared in the cases of 4x typical haptotaxis strength (**Fig 9B**) and 4x typical VEGF initial concentration (**Fig 9C**).

**Fig 10** shows the time history of the junction number change. We multiplied each typical parametric value to investigate the parameter effect involved. We combined the VEGF binding and unbinding rates as one parameter $k_{on}/k_{off}$ because they share virtually the same curve with

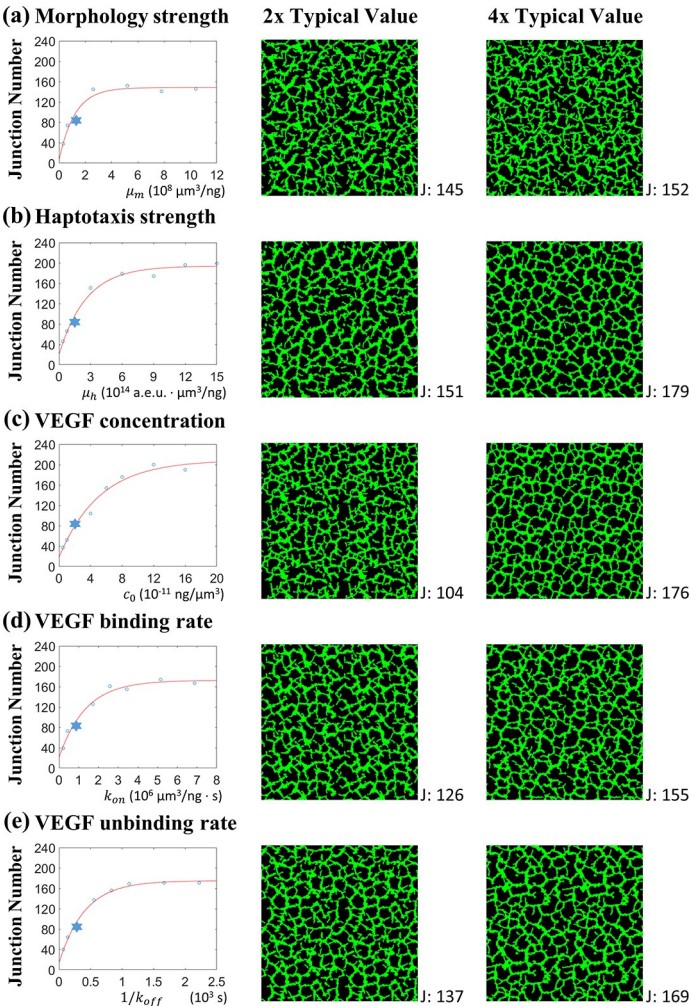

**Fig 9. Network junction number at 24 hours of incubations for various values of VEGF parameters.** In each set, one parameter changes as denoted in the abscissa, and the other parameters keep their typical values as in **Table 1**. The asterisk denotes the junction number at the typical parameter values of **Table 1**. The network pattern for the 2x and 4x corresponding typical parameter values is shown on the right. The junction number in the network is denoted beside the network. A more complete network is of a more significant junction number, which approaches the asymptotic value as the parameter value increases, and there is virtually no open loop in sight when the junction number exceeds 170.

similar intercepts, slopes, and plateau values (**Fig 9D and 9E**). The regression curve $f(t)$ proposed based on the observed data trends reads the following form

$$f(t) = A\left(1 - e^{-B\mu_m - C\mu_h - Dc_0 - Ek_{on}/k_{off}}\right)\left[1 - e^{-F \cdot \mu_m{}^G \cdot \mu_h{}^H \cdot c_0{}^I \cdot (k_{on}/k_{off})^J \cdot t}\right] \tag{2}$$

**Table 2. Curve fitting coefficients for Eq (1).**

| $P$ | $a$ | $b$ | $c$ |
|---|---|---|---|
| $\mu_m$ | 148 | 143 | $8.32 \times 10^{-9}$ ng/μm$^3$ |
| $\mu_h$ | 194 | 174 | $3.87 \times 10^{-15}$ ng/μm$^3$ a.e.u. |
| $c_0$ | 209 | 191 | $1.95 \times 10^{10}$ μm$^3$/ng |
| $k_{on}$ | 173 | 151 | $7.38 \times 10^{-7}$ sec ng/μm$^3$ |
| $1/k_{off}$ | 175 | 161 | $2.45 \times 10^{-3}$ sec$^{-1}$ |

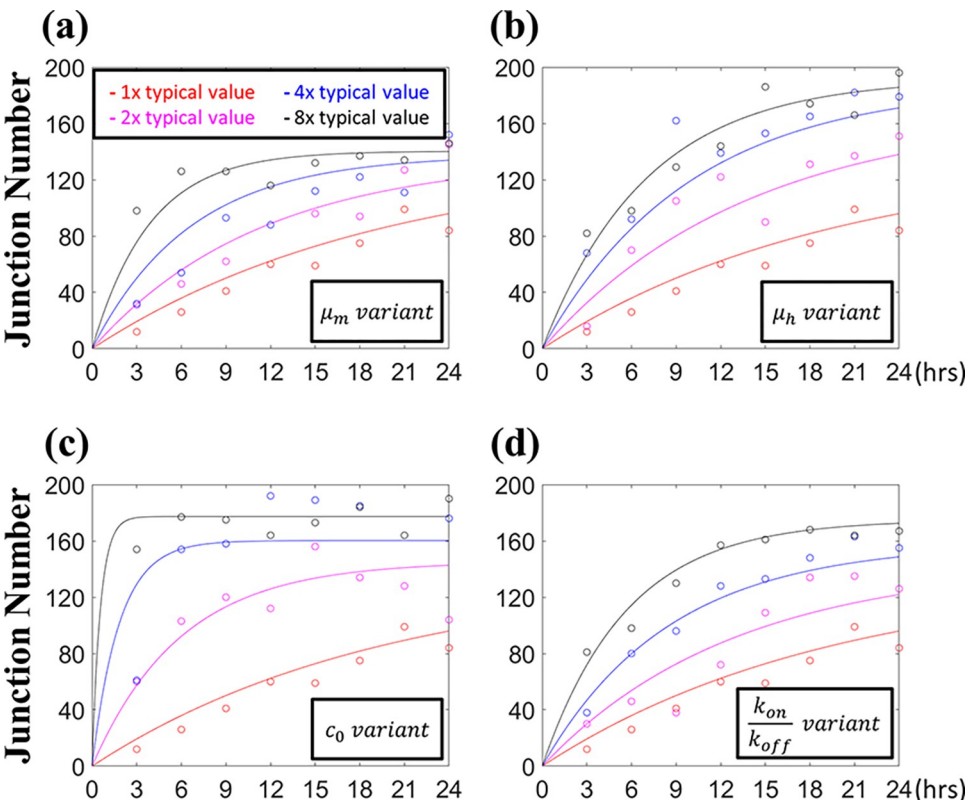

**Fig 10. Temporal evolution of the junction number for different cell migration parameters.** The results show that increasing these four parameter values raised the junction number in stable cellular networks. Increasing the initial VEGF concentration increased the junction number more significantly than the other parameters. In each figure, all the parameters are as the typical values in **Table 1**, except for the parameter in the box, which is adjusted by multiplying its typical value.

The fitted values of coefficients are listed in **Table 3** with the coefficient of determination $r^2$ = 0.9472 and adjusted coefficient of determination $r^2$ = 0.9447. The first exponential term denotes the junction number of the final stage, and the regressions suggested that increasing the haptotaxis strength $\mu_h$ and the soluble VEGF amount $c_0$ raised the junction number more than expanding the morphology strength $\mu_m$ under the same magnification rates. The term in the square bracket accounts for the time rate of the network development. The rising speed of the junction number was higher if we multiplied the initial soluble VEGF concentration $c_0$. Indeed, increasing the initial VEGF amount was equivalent to increasing both the soluble and bound VEGF concentration contemporarily, so it played a more significant effect than other parameters.

**Table 3. Curve fitting coefficients for Eq (2).**

| | | | |
|---|---|---|---|
| $A$ | 192 | $F$ | $1.64 \times 10^{-5}$ (ng/µm$^3$)$^{0.152}$ a.e.u.$^{0.489}$ sec |
| $B$ | $1.03 \times 10^{-10}$ ng/µm$^3$ | $G$ | 0.777 |
| $C$ | $5.60 \times 10^{-15}$ ng/µm$^3$ a.e.u. | $H$ | 0.489 |
| $D$ | $9.60 \times 10^{9}$ µm$^3$/ng | $I$ | 1.707 |
| $E$ | $7.20 \times 10^{-10}$ ng/µm$^3$ | $J$ | 0.593 |

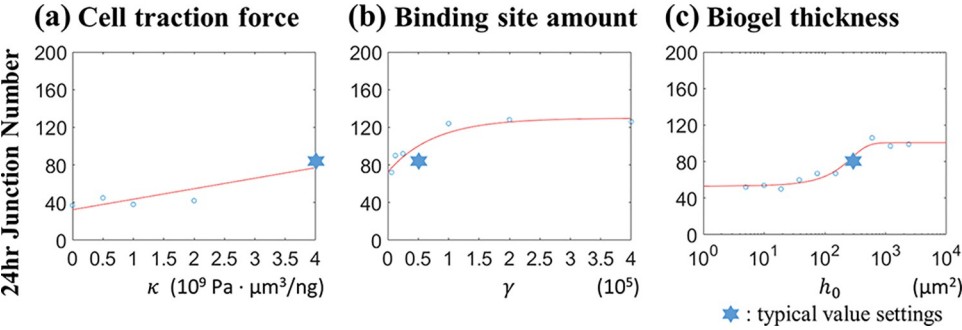

**Fig 11. Relationships of the junction number at 24 hours of incubations with biogel parameters.** The regression curves that fit the data trend depend on the data distribution within the simulation intervals. In each figure, one parameter changes as denoted in the abscissa, and the other parameters keep the typical values as in **Table 1**.

## 2.5. Biogel parameter analysis

As for the parameters related to biogel deformation in the simulation, we focused on interactive parameters for the three parts: cells, biogel, and VEGFs. The first was the strength of cell traction force $\kappa$, which regulates the biogel responses from cell adhesion. The second was the ratio of VEGF binding sites $\gamma$, which bridges the communications between the VEGF and biogel. Finally, the biogel thickness $h_0$ was also considered for verification purposes. Other parameters related to biogel properties were not changed in value in investigations.

**Fig 11** shows the junction number at 24 hours. We changed these parameters around their typical values and performed the regression fits based on the data trends. The regression equation relating the junction number with the strength of cell traction force $\kappa$ is

$$f(\kappa) = a_1 + b_1\kappa \tag{3}$$

The regression equation for the junction number with the ratio of VEGF binding sites $\gamma$ is

$$f(\gamma) = a_2 - b_2 e^{-c_2\gamma} \tag{4}$$

The last set of data points for the various biogel thicknesses yields the logistic function as follows

$$f(h_0) = a_3 + \frac{b_3}{1 + e^{-c_3 h_0 + d_3}} \tag{5}$$

The fitted values of coefficients are listed in **Table 4**. The results showed positive correlations between the network junction and the three parameters. Eq (3) reveals the greater traction force resulted in higher biogel deformation. Eq (4) shows that increasing the VEGF binding sites led to increased bound VEGF accumulation for inducing cell migration. Eq (5) presents that the predicted junction number approached an asymptotic value as the gel increased in thickness, for the upper surface of the thicker biogel received less restriction from the Petri dish. All these results depicted the correlation between the biogel deformation and

**Table 4. Curve fitting coefficients for Eq (3)–(5).**

| $n$ | $a_n$ | $b_n$ | $c_n$ | $d_n$ |
|---|---|---|---|---|
| 1 | 32.5 | 1.12 x $10^{-8}$ ng/$\mu$m$^3$ Pa | - | - |
| 2 | 129.8 | 57.4 | 1.31 x $10^5$ | - |
| 3 | 36.6 | 64.0 | 7.72 x $10^{-3}$ $\mu$m$^{-1}$ | 1.1 |

the cellular network formation, which agreed with the experimental observations [43]. In summary, the capillary-like network development relied on adequate biogel deformation, a sufficient amount of bound VEGF, and the gradient of bound VEGF to offer spatial cues for the endothelial cells. The cellular network could be well-developed only when the mechanical and chemical coupling worked simultaneously.

## 3. Discussion

Our simulations showed that the cellular-like network was well-developed if the mechanical and chemical coupling mechanisms could work simultaneously. We compare the cellular networks between the literature experiment [44] and the current simulation at 18 hours (**Fig 1**), even though we have collected the data until 24 hours. The experiment showed no significant drop in the number of living cells before the last 6 hours, supporting the apoptosis-neglected assumption for simulation, which is valid till 18 hours. Literature has shown that after 18 hours of incubation, the hollowed spaces inside the cellular cords could be observed [7,43]. We thus regarded the stable capillary-like network at 18 hours as our simulation target. The agreement between simulation and experiment results presents the reliability of our mathematical model. Our results successfully simulated the formation of cellular networks and the co-existence of cobblestone alliances observed in previous experiments [7,44,56]. Based on experimental observation, we suppressed the cell elongation if it was surrounded by other cells in the simulation. Detailed mechanisms behind this phenomenon require further investigation.

The morphologized endothelial cells from previous studies showed the following features: elongation in shape, cytoskeletal solidification, and cytoplasm merging between connected cells [46]. The arching of the elongated cells benefited the subsequent cord hollowing, so a slight decrease in the occupied top-view area was observed under the phase-contrast microscope [46,52]. Although our modeling only involved the elongation probability for each cell, the solidification of the cell cytoskeleton was partially presented even in the current 2-D simulation. Our cellular shape constraints for area and perimeter were balanced in the CPM. Thus, smaller cells were obtained compared to the original. However, Our simulation ignored the cytoplasm merging between the cells. This stabilization issue was negligible in normal conditions when cells gathered due to haptotaxis but would lead to mismatches if the haptotaxis was absent. Repeated attachment and detachment of elongated cells were observed in the simulation case of morphology-on and haptotaxis-off (**Fig 3C**). In contrast, an incomplete but maintainable network in the experiments without haptotaxis driven by bound VEGF was shown since the cytoplasm merge stabilized the connected cells [44].

The present model focused on the cell elongation cued by the soluble VEGF without considering the detailed intracellular events activated by the soluble VEGF. The chemical-based model of Scianna and Munaron (2011) considered a series of intracellular events, including activating protein kinase and releasing arachidonic acid, nitric oxide, and calcium ions [34]. Their model may provide a basis for further investigating the intracellular signal path for cell elongation cued by soluble VEGF. The VEGF secretion from and later consumption by endothelial cells was also included in their model, which resulted in a non-constant total VEGF amount. The VEGF consumption could decrease the VEGF concentration over time, reducing cellular morphology and haptotaxis tendencies, and might cause an earlier but less complete network [42].

Our model considered the bound VEGF was accumulated along with the biogel around the cells through the cell traction, which is consistent with the finding of Malandrino et al. (2019) [57]. They showed that cells can accumulate ECM around them through pulling. In our

simulation, the concentrated bound VEGF under the cells drove the haptotaxis and the alignment of the elongated cells. However, the present model has ignored the transport of soluble VEGF hindered by the attached cells on the substrate. The VEGF diffusion and binding and unbinding rates on the cell-attached surface were as if on a cell-free surface. Such hampering of the VEGF diffusion may cause the VEGF concentration in the biogel surface to be lower in regions with attached cells and indirect contact with the medium. It may delay and hinder the formation of a cellular network, which should be addressed in future work.

In the era when VEGF functions were unclear, theoretical works assumed that cellular networks developed purely due to the biogel deformation caused by the traction of endothelial cells without including bound VEGF [27–29]. Such theoretical approaches obtained similar co-localized cells and biogel (**Fig 2**). We also considered gel deformation under the friction between the gel and the Petri dish so that our results could better explain the tilted biogel experiments [43]. Comparing our results to the more recent mechanical-based model from van Oers et al. [37], we neglected the effects of stress gradients inside the cells. van Oers et al. considered that the intense substrate strain directly guides cell movement and deformation. They showed this mechanism can reproduce network formation from scattered cells and sprouting from cell spheroids. Since we studied cell morphology change signaled by soluble VEGF, we thus neglected substrate stress gradients inside the cell regions. The substrate deformation in the present model induced the bound VEGF gradients, which guided the cell directional migration. The soluble and bound VEGF effects on cell morphology and migration should be regarded as a feature for endothelial cells rather than for all the attached-type cell lines [58]. The bound VEGF was consensusly regarded as dominating cell migration and network formation in the chemical-stimulation part. The oriented cue was sourced from the non-uniform distributed binding on biogel [35,42,59] by further incorporating the soluble VEGF in the model.

## 4. Conclusion

We have developed a comprehensive model encompassing mechanical regulation and chemical stimulation cues for forming capillary-like networks. The model covers the respective mechanisms of two VEGF forms on endothelial cells. The soluble VEGF in the culture medium induces morphological changes, and the bound VEGF on the biogel surface provides oriental signals, both playing roles in forming the capillary-like networks. The simulation successfully demonstrated that cells gathered from scattered individuals to cell cords and then from the cell cords to an overall network. The results were consistent with the experiments, showing that the cellular network was well-developed if the mechanical and chemical coupling worked simultaneously. The current model is valid for short-term static incubation. Furthermore, for the practical transplantation into living bodies to promote neovascularization around the surgical site [18,51,60], an extension of our model may cover the extracellular remodeling and endothelial cell morphology in the future.

## 5. Methods

The theoretical formulation depicts the *in vitro* capillary network development, considering the interactions among the endothelial cells, the biogel substrate, and the VEGF in soluble and bound forms. A two-dimensional hybrid model accounts for the endothelial cell adhesion and migration on the biogel surface, neglecting the cell penetration into the gel [43,61]. As the schematic diagram shown in **Fig 12**, The model considers the biogel deformation due to traction by the attached cells. The un-uniform distribution of bound VEGF due to the gel deformation offers a directional migration cue for the cells via haptotaxis. The soluble VEGF

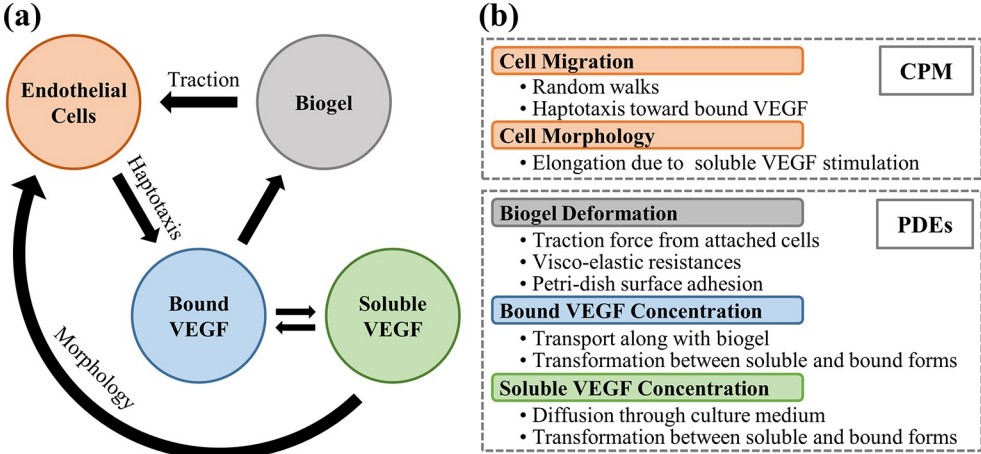

**Fig 12. Schematic diagram of (a) the cells-VEGF-biogel interplay and (b) the mechanisms included in the CPM and PDEs model.** The cells exert traction force on substrates, causing the biogel deformation and non-uniform bound VEGF distribution. The bound VEGF gradient signal directed cell migration, resulting in a positive loop for cell aggregation, biogel deformation, and bound VEGF local accumulation. The two forms of VEGF transform dynamically. The soluble VEGF stimulates endothelial cells to initiate the elongated morphological change.

activates the morphology change so that the cells elongate in shape. As mentioned above, our mechanical mechanism is not direct between the cells and substrate. Still, it refers to the traction of biogel by cells, which provides a haptotaxis cue through the bound VEGF. The cell-based CPM predicts cellular individual behaviors, including migration, shape deformation, and morphology change [31,32]. The continuum-based PDEs formulate the temporal and spatial distributions of biogel and VEGFs [33–35,42].

## 5.1. Cellular Potts model

Cell proliferation and apoptosis are negligible for the simulation of short-term incubation [5,6]. Based on the two-dimensional assumption, the hollowing structure in cell morphology is also neglected in our model [52]. In CPM, each cell occupies several connected Cartesian lattices, and a hypothetical energy of the cell system is defined using the following Hamiltonian [31,33]

$$H = \sum_{\{x.x'\}} J_{\tau_x \tau_{x'}}(1 - \delta_{\sigma_x \sigma_{x'}}) + \sum_{\sigma > 0}(\lambda_a(a_\sigma - a_0)^2 + \lambda_p p_\sigma + \lambda_l \phi_\sigma / l_\sigma) \quad (6)$$

The variable $x$ presents a specific lattice site and $x'$ its neighboring lattices. We considered the first- and second-order neighborhood in the present modeling (**Fig 13**). The subindex $\tau_x = 0, 1$ in Eq (6) denotes whether the lattice $x$ is cell-free or cell-occupied, respectively. The subindex $\sigma_x = n$, a non-negative integer, represents the lattice precisely occupied by the $n$-th cell and $\sigma_x = 0$ if the lattice is cell-free. The first term in Eq (6) describes the interface energy summation of the cell system, in which the interface energy $J_{\tau_x \tau_{x'}}$ between the two neighboring lattices takes different values depending on the neighboring states, and a Kroneckor delta formulation $(1 - \delta_{\sigma_x, \sigma_{x'}})$ excludes counting the interface energy for the two neighboring lattices belonging to the same cell. The second-order neighbors contribute less energy with the weight $1/\sqrt{2}$ than the first-order neighbors, considering their distance is $\sqrt{2}$ times longer.

The second term in Eq (6) accounts for the cell deformation limitations, where $a_\sigma$, $p_\sigma$ and $l_\sigma$ are the cell area, perimeter, and head-to-tail length, respectively, and $\lambda_a$, $\lambda_p$, $\lambda_l$ the

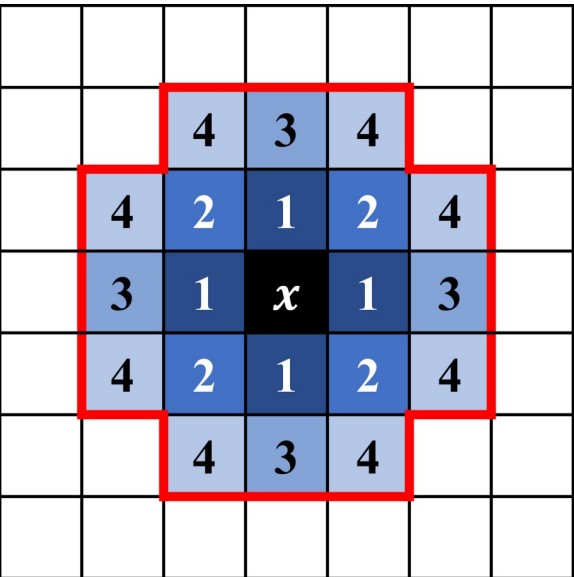

**Fig 13. Initial cell shape and the n-th order neighborhoods in CPM.** An individual cell occupies 21 connected lattices, initially approximating a round shape. The marked center lattice demonstrates four different types of neighboring lattices. The higher-order neighboring lattices are less likely to be selected as the extended site $x$' because they are far from the lattice $x$. We adopted the first and second-order neighborhoods in our simulation. The second-order neighboring lattices are $1/\sqrt{2}$ less likely to be selected than the first-order ones.

corresponding restraint coefficients. The first and second restraints keep the initial cell area $a_0$ and maintain a minimal peripheral size to prevent the cell from breaking into pieces [31,42]. The third restraint guides the endothelial cells to undergo cytoskeletal remodeling by changing their shape from round to elongated. The index $\phi_\sigma = 0,1$ labels whether the cell is still in its initial flattened state ($\phi_\sigma = 0$) or has changed its morphology ($\phi_\sigma = 1$). Once the cell has activated its shape remodeling, it is keen to increase its length $l_\sigma$, presenting the differentiation for endothelial cells as irreversible [54].

The cell migration simulation followed the modified Metropolis algorithm [62]. First, a lattice $x$ in the computational domain and one of its neighbors $x$' were chosen, where the picking process was random and independent of the current lattice state. The state of the lattice $x$ was assumed to extend to the lattice $x$' and thus caused the energy variation $\Delta H$ in the cell system. The Hamiltonian variation was further modified to include $\Delta H_m$ for the haptotaxis effect by considering the bound VEGF gradient [24,25]

$$\Delta H_m = \Delta H - \mu_h(b_{x'} - b_x)(1 - \delta_{\tau_x \tau_{x'}}) \tag{7}$$

The coefficient $\mu_h$ represents the haptotaxis strength, and the $b$'s are the bound VEGF concentrations at the two lattices. The term $(1 - \delta_{\tau_x \tau_{x'}})$ excludes haptotactic migration if both sites are occupied by cells, accounting for the phenomenon of cell contact inhibition [63]. Whether or not the displacement happens was determined by the following probability [31]

$$P(\text{displacement}) = \begin{cases} 1 & , \ \Delta H_m \leq 0 \\ e^{-\Delta H_m}, & \text{otherwise} \end{cases} \tag{8}$$

The Potts model commonly takes the form $\exp(-\Delta H_m/T)$, including an additional temperature parameter $T$ to quantify the magnitude fluctuation [31]. We kept the temperature

parameter constant without considering the fluctuation effect. A random number between 0 and 1 was generated and compared with the probability to decide whether the state change at x' truly happens. Eq (8) allows a possible move even under energy-raising conditions, representing the random walks for living cells. Finally, by repeating the above steps until the number of repetitions reaches the total lattice number, each lattice would have an equal opportunity for its state update, and the system would go through a complete Monte-Carlo step (MCS) equivalently [64].

The current model assumed all the cells were initially round-shaped. The possibility of the cell morphological change at each time step depended on the local concentration of soluble VEGF, which takes the following form.

$$P(\text{morphological change}) = \mu_m c_\sigma r_\sigma \tag{9}$$

where $\mu_m$ is the activation coefficient, $c_\sigma$ the soluble VEGF concentration averaged over the identical cell-occupied lattices, and $r_\sigma$ the no-cell-contact interface ratio to its total perimeter. As reported in literature experiments, we included $r_\sigma$ keeping the cell in a cobblestone-like group to maintain its round, flattened shape [7,44,45].

## 5.2. Biogel PDE model

We neglected the secretion of extracellular matrix (ECM) by the endothelial cells, the limited enzymatic degradation rate of biogel, and the biogel proteolysis, considering the short-term incubation [7,65]. Therefore, the time change rate of the local gel density $\rho$ due to the biogel displacement is described by the conservation equation [27]

$$\frac{\partial \rho}{\partial t} = -\nabla \cdot \left( \rho \frac{\partial \boldsymbol{u}}{\partial t} \right) \tag{10}$$

where $\boldsymbol{u}$ is the biogel displacement vector. The biogel is considered a viscoelastic substrate. By ignoring the inertia force compared to the visco-elastic force, we adopted the following force-balanced equation to evaluate the biogel displacement [29]

$$\nabla \cdot (\boldsymbol{\sigma}_{cell} + \boldsymbol{\sigma}_{vis} + \boldsymbol{\sigma}_{ela,linear} + \boldsymbol{\sigma}_{ela,long-range}) = \boldsymbol{R}_{ext} \tag{11}$$

The first term $\boldsymbol{\sigma}_{cell}$ represents the cell traction stress exerted on the surface, which is assumed proportional to the biogel density and formulated as

$$\boldsymbol{\sigma}_{cell} = \kappa \rho \tau_x \boldsymbol{I} \tag{12}$$

where the parameter $\kappa$ regulates the traction amplitude, the index $\tau_x = 0$, 1 coupled with the CPM indicates the stress existing only at the cell locations and $\boldsymbol{I}$ is the identity tensor. The second term $\boldsymbol{\sigma}_{vis}$ in Eq (11) represents the viscous stress of the biogel. According to the Stokes law of deformation, the viscous stress is modeled as [27]

$$\boldsymbol{\sigma}_{vis} = \mu_1 \frac{\partial \boldsymbol{\varepsilon}}{\partial t} + \mu_2 \frac{\partial \theta}{\partial t} \boldsymbol{I} \tag{13}$$

Eq (13) assumes the viscous stress is proportional to the biogel strain and dilatation rates. In the equation $\boldsymbol{\varepsilon} = (\nabla \boldsymbol{u} + (\nabla \boldsymbol{u})^T)/2$ is the gel strain, $\theta = \nabla \cdot \boldsymbol{u}$ the gel dilatation, $\mu_1$ and $\mu_2$ the shear and bulk viscosities, respectively. The terms $\boldsymbol{\sigma}_{ela,linear}$ and $\boldsymbol{\sigma}_{ela,long-range}$ in Eq (11) are the local and long-range linear elastic stresses, respectively. In addition to the local stress, we assume that the elastic force includes that from the far-apart surroundings and is transmitted via the fiber components in the biogel. The local linear elastic stress takes the following form

following Hooke's law [27]

$$\boldsymbol{\sigma}_{ela,linear} = \frac{E}{1+\upsilon}\left[\boldsymbol{\varepsilon} + \frac{\upsilon}{1-2\upsilon}\theta\boldsymbol{I}\right] \quad (14)$$

where $E$ is the Young's modulus and $\upsilon$ is the Poisson ratio of the biogel. The long-range elastic stress is modeled with the Laplacian terms for its ranged effects [29]:

$$\boldsymbol{\sigma}_{ela,long-range} = \frac{E}{1+\upsilon}\left[-\beta_1\nabla^2\boldsymbol{\varepsilon} - \frac{\upsilon}{1-2\upsilon}\beta_2\nabla^2\theta\boldsymbol{I}\right] \quad (15)$$

where $\beta_1$ and $\beta_2$ are the non-negative long-range elastic coefficients. The last term $\boldsymbol{R}_{ext}$ in Eq (11) represents the friction force on the biogel by the Petri dish. As detailed in **S1 Appendix**, the attachment effect is expressed by the following form

$$\boldsymbol{R}_{ext} = \frac{1}{h^2}\left[\frac{E}{1+\upsilon}\left(\boldsymbol{u} - \beta_1\nabla^2\boldsymbol{u}\right) + \mu_1\frac{\partial\boldsymbol{u}}{\partial t}\right] \quad (16)$$

where $h$ is the biogel thickness, which depends on the local gel density $\rho$, the initial thickness $h_0$, and the initial density $\rho_0$

$$h = h_0\left[\frac{1-3\upsilon}{1-2\upsilon} + \frac{\upsilon}{1-2\upsilon}\frac{\rho}{\rho_0}\right] \quad (17)$$

## 5.3. VEGFs PDE model

In general, the VEGF dynamics in cellular incubation systems contain the secretion from specific cell types, cell consumption via pinocytosis, and molecular degradation in the culture medium [23,42]. For simplicity, we assumed the total amount of VEGF was a constant for simulating the short-term cultivation in this work. We considered two VEGF forms in the current model. The soluble form, the solute VEGF in the culture medium, can stimulate the cell morphological remodeling of elongation in cell shape. The bound form, the VEGF binding to biogel, can induce cell migration toward the haptotaxis cues [24,25,44]. The two VEGF forms transform mutually via binding sites such as fibronectin and heparin sulfates in the biogel. Following the experimental designs in the literature [25,65], we presumed a finite amount of fibronectin in the biogel. The dynamic balance between the local concentrations of the soluble VEGF $c$ and the bound VEGF $b$ were formulated as

$$\frac{\partial c}{\partial t} = D_c\nabla^2 c - k_{on}c\left(\gamma\rho - \frac{M_{FN}}{M_{VEGF}}b\right) + k_{off}b \quad (18)$$

$$\frac{\partial b}{\partial t} = -\nabla\cdot\left(b\frac{\partial\boldsymbol{u}}{\partial t}\right) + k_{on}c\left(\gamma\rho - \frac{M_{FN}}{M_{VEGF}}b\right) - k_{off}b \quad (19)$$

where $D_c$ is the diffusion coefficient of soluble VEGF, $k_{on}$ and $k_{off}$ respectively represent the binding and unbinding rates of the VEGFs, and $\gamma$ is the content ratio of VEGF binding sites to the biogel, wherein we assumed one fibronectin molecule provides a binding site [14]. Finally, the term $(\gamma\rho - bM_{FN}/M_{VEGF})$ evaluates the concentration of the free binding sites that remain conformed to the aforementioned binding constraint by including the molecular weight ratio of fibronectin $M_{FN}$ to VEGF $M_{VEGF}$.

### 5.4. Numerical simulation

The hybrid model includes the CPM and PDEs. In-house coding with MATLAB was for CPM computation. COMSOL Multiphysics was adopted to solve the PDEs. The computational domain spanned 170x170 lattices. We set the squared lattice as 10 μm in width and MCS as 2 minutes so that the geometric center of each cell was monitored migrating with a maximum 30 μm/hr speed [66]. Initially, a cell in a virtually round shape occupied 21 connected lattices (**Fig 13**). At the beginning of the simulation, 400 cells spread randomly in the lattice grid so that the number density for living cells was spatially comparable with the experimental reference [44]. The grid size for the PDE models was identical to that of the CPM lattice. The biogel density and soluble VEGF concentration initially had a uniform distribution, whereas the initial concentration of bound VEGF was zero. Periodic conditions were specified at the four boundaries for the CPM and PDEs. These boundaries allow cells to migrate across and show up on the opposite side equivalently.

**Table 1** presents the model parameters and their typical values. The endothelial cell line in the simulation referred to the human umbilical vein endothelial cells (HUVECs), and the primary biogel composition was the air-dried type I collagen. We determined the "This work" values in **Table 1** by fitting the cell patterns with previous experiment data [44]. The morphology coefficient $\mu_m$ was first determined to have approximately one-third of the cells become elongated after 12 hours of incubation, which was the observed amount from the reference experiments [44]. The cell-shape-related parameters $\lambda_a$ and $\lambda_p$ were then determined based on the round cell features of approximately 50 μm in diameter and 1960 μm$^2$ in area, and $\lambda_l$ was determined based on the elongated cell feature of 2.5 times the initial diameter [44]. For the biogel parameters, the traction amplitude $\kappa$ was determined so that the biogel experienced deformation comparable with previous simulation works [29,59], and the amount of fibronectin was chosen via the fibronectin ratio $\gamma$, which allowed approximately 10% of bound VEGF to transform from the soluble form [67]. Finally, the haptotaxis coefficient $\mu_h$ was determined so that the overall cell pattern approached a capillary-like network after 18 hours of incubation, as shown in the experiments [7,43,44].

### 5.5. Images and statistical analysis

In this work, the number of cellular network junctions, segments, and the total length of the segments in the computational domain were adopted as indicators to quantify the cellular network integrity and complexity [14,68]. A 100 μm threshold was applied to identify the network segments, and the network nodes intersected by at least three segments were judged as junctions. We further performed the cell shape analysis with the roundness index defined as:

$$\text{roundness} = \frac{4a_\sigma}{\pi p_\sigma^2} \tag{20}$$

The dip unimodal test was performed for the cell migration trends [53]. We adopted MATLAB R2020b and COMSOL Multiphysics 5.6 for numerical simulations, Image J 1.52a with the Angiogenesis Analyzer plugin for the network completeness check and the cell shape studies, and Microsoft Excel 2016 for Student's t-test statistical analysis. The regression analysis for the junction number and the dip tests were also performed using MATLAB R2020b.

## Supporting information

**S1 Appendix. Friction Force Exerted by the Petri Dish.**
(DOCX)

## Author Contributions

**Conceptualization:** Hsun Chiang, Chih-Ang Chung.

**Data curation:** Hsun Chiang.

**Formal analysis:** Hsun Chiang, Chih-Ang Chung.

**Funding acquisition:** Chih-Ang Chung.

**Investigation:** Hsun Chiang.

**Methodology:** Hsun Chiang, Chih-Ang Chung.

**Project administration:** Chih-Ang Chung.

**Resources:** Hsun Chiang, Chih-Ang Chung.

**Software:** Hsun Chiang.

**Supervision:** Chih-Ang Chung.

**Visualization:** Hsun Chiang.

**Writing – original draft:** Hsun Chiang.

**Writing – review & editing:** Hsun Chiang, Chih-Ang Chung.

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
