## [Decision Letter · Decision Letter 0]

12 Feb 2024

Dear Prof Chung,

Thank you very much for submitting your manuscript "Simulation of Soluble and Bound VEGF-stimulated in vitro Capillary-like Network Formation on Deformed Substrate" for consideration at PLOS Computational Biology.

As with all papers reviewed by the journal, your manuscript was reviewed by members of the editorial board and by several independent reviewers. In light of the reviews (below this email), we would like to invite the resubmission of a significantly-revised version that takes into account the reviewers' comments.

We cannot make any decision about publication until we have seen the revised manuscript and your response to the reviewers' comments. Your revised manuscript is also likely to be sent to reviewers for further evaluation.

Sincerely,

Roeland M.H. Merks, Ph.D

Academic Editor

PLOS Computational Biology

Jason Haugh

Section Editor

PLOS Computational Biology

Reviewer's Responses to Questions

**Comments to the Authors:**

Reviewer #1: The authors use cellular Potts model simulations to investigate how soluble and bound VEGF signaling as well as mechanical stimuli from a biogel affect the developing morphology of blood vessels.

The authors provide a good summary of previous work on modeling blood vessels - I found this very helpful. They explain their model well, and provide a good table of parameters.

Overall, this is an interesting paper that merits publication after some issues are addressed.

Major comments:

(A) First, the authors say that they shared the implementation and simulation data on Zenodo, but it seems that the repository is not set up properly, so it was not possible to access that data or code.

(B) Section 2.1 is a bit too much CPM review and detail. I recommend to put most of this in an Appendix, as readers of PLOS CB will have seen such material often, and to keep in the main text only the distinct aspects used for this paper. (For example, Eq. 3, etc). I also suggest to eliminate Figure 2 or put it into an Appendix.

(C) The figure captions are very brief. They should contain more detail, so that they basically give the narrative of the paper.

(D) If I understand it correctly, the mechanical stress in the biogel is only transmitted to the cell through the intermediate effect of the bound VEGF (in the effective force in Eq. 3), not directly. If so, this should be justified. If not, this possible confusion should be clarified. Is this a limitation of the model?

(E) In Eq (4), the authors left out the usual fluctuations parameter ("temperature"). Did they consider varying this, and how would it affect the results?

(F) Fig 6, the authors state that each numerical simulation was repeated three times. Is there a reason to think that this is enough to calculate accurate statistics?

(G) To compare the topology of the simulated network with real data, the authors chose various properties, such as number of junctions, number of segments, total length of segments. Why were those chosen as best metrics, rather than other measures (like vessel diameter distribution, loop or cycle analysis, fractal dimension, density, pore size, etc.)

(H) There are two papers whose model and results should also be included in the review of previous work, and compared with the current paper. There are some similarities.

[1] Scianna, Marco, and Luca Munaron. "Multiscale model of tumor-derived capillary-like network formation." Networks Heterog. media 6.4 (2011): 597-624.

[2] van Oers, R. F., Rens, E. G., LaValley, D. J., Reinhart-King, C. A., & Merks, R. M. (2014). Mechanical cell-matrix feedback explains pairwise and collective endothelial cell behavior in vitro. PLoS computational biology, 10(8), e1003774.

Minor comments:

L134 VEGF "solutal"  should it be "soluble"?

(Appears also on L242)

L 269 What is meant by the "passable condition"

Reviewer #2: Title: Simulation of Soluble and Bound VEGF-stimulated in vitro Capillary-like Network Formation on Deformed Substrate.

The manuscript by Chiang and Chung presents a hybrid model for short-term in vitro simulation of capillary networks. The authors use the Cellular Potts Model (CPM) to represent individual cell migration, and employ partial differential equations (PDEs) to quantify biogel deformation and vascular endothelial growth factor (VEGF) transport dynamics. A key aspect of their mathematical model is the inclusion of both soluble and bound forms of VEGF. The model proposes that cell morphology is influenced by soluble VEGF, while haptotaxis is driven by bound VEGF. The authors effectively demonstrate that the inclusion of these two forms of VEGF allows their model to form capillary-like networks that are consistent with experimental observations. They further investigate the characteristics of these networks in terms of the number of junctions, number of segments, and total segment length, revealing the crucial roles of soluble and bound VEGFs in the network formation process.

Overall, the demonstration that both forms of VEGF are essential for the formation of capillary-like networks is an important contribution. I would therefore support publication, subject to the following comments:

Major comments:

#1: In section 2.1, the authors provide a description of the Cellular Potts Model (CPM). While the explanation is thorough, the addition of a schematic diagram to visually represent the model would greatly enhance comprehension, particularly for readers who may not be theoreticians. Given the diverse audience of this journal, including life scientists, a diagram could facilitate a more intuitive understanding of the CPM model and how it applies to your study.

#2: In Figure 6, the features of the network are assessed using junction number, segment number, and total segment length for each of the four conditions. However, I find that the definitions of these metrics are not fully clear. Could you provide further clarification on what exactly is meant by each of these terms? A diagram or a brief explanation in the manuscript would be helpful for readers to understand how these metrics contribute to the characterization of the network.

#3: The authors' CPM shows that a more realistic network is formed by the cellular cords composed of elongated cells and a cobblestone-like aggregate of round cells. In this context, how does the distribution of cell shapes, such as aspect ratio and roundness, change under the four conditions of morphology and haptotaxis on or off? I am particularly curious to see how turning haptotaxis on or off affects these shape distributions.

#4: In Figure 6, the network characteristics are examined under four conditions - with morphology and haptotaxis either on or off - using three metrics. The manuscript suggests that the presence or absence of haptotaxis affects cell distribution, leading to more random patterns. This makes me interested in how these differences can be assessed quantitatively. Are there specific metrics or approaches that could be used to clearly delineate network characteristics, particularly in terms of topology, under different haptotaxis conditions?

#5: Regarding the fitting curves chosen for the parameter analysis in sections 3.4 and 3.5, could you clarify whether they were chosen on the basis of statistical theory or whether they were determined empirically on the basis of the observed data patterns? A brief explanation of the rationale for the choice of these fitting curves would enhance the understanding of the authors' analytical approach.

Minor comments:

#6: I noticed that Figure 3(b) shows a snapshot of the simulation, but the caption does not specify the time after which this snapshot was taken. Could you clarify in the caption the specific time represented in this snapshot? For example, does this image show the state of the simulation 18 hours after it was started?

#7: I noticed that Figure 6(b) contains several bar graphs, but there is no description in the caption of the error bars shown on these graphs. Do they represent the standard error (SE) or the standard deviation (SD)?

#8: I would like to draw your attention to a couple of minor typographical errors for correction. First, in line 72, it might be helpful to add the abbreviation '(PDEs)' after the first mention of 'partial differential equations' for clarity. Also, in line 126, 'VEG' seems to be a typo, and I think it should be corrected to 'VEGF'.

Reviewer #3: In “Simulation of soluble and bound VEGF-stimulated in vitro capillary-like network formation on deformed substrate” the authors implement a Cellular Potts model of vasculogenesis and carry out a quantitative comparison of their results with experiments. This comparison is a strong point of the manuscript. However, I am of the opinion that the novelty of the modelling implemented is not sufficient to warrant publication in PLoS Computational Biology.

Major Points:

=> line 171. The authors introduce the probability for the cell to become elongated per unit time. Does it become elongated permanently? If the cell moves to a region with a low VEGF concentration, or if it becomes surrounded by other cells, can it transform back to the flattened phenotype? This dynamics of phenotype change is not well explained in the document. Why is the value of \\mu_m so large? It seems that immediately that the cell can, it will change to its elongated phenotype. Also, though I understand the reason for introducing r_sigma, so that when a cell is surrounded by other cells it does not become elongated, it is very hard to understand from the biological point of view. A cell should have VEGF receptors on all its membrane surface, including the part above (the apical region), and the soluble VEGF should bind those receptors, not only the receptors at the perimeter of the cell. Therefore, the presence of neighbouring cells should not affect significantly the ability for a cell to bind soluble VEGF when it is plated in a flat substrate.

=> line 215. Here, the stress is approximately constant throughout the cell. That is a different approach from van Oers et al 2014, and from the experimental measures of traction force microscopy by Reinhart-King group, where the traction force increases from the center to the periphery of the cell. Why did the authors decided to implement a stress that is approximately constant inside the cell?

=> lines 249, 250. The authors should justify why VEGF intake by the endothelial cells is not considered.

=> The process of choosing the parameters is not conveniently explained:

=>=> MCS is chosen as 2 minutes so that the cells migrate at a “reasonable speed”. What is a reasonable speed? Did the authors carry out migration simulations of isolated cells? How was the comparison with experiment carried out?

=>=> mu_m was determined to have approximately one third of elongated cells? Why one third? How different is the number of elongated cells in the simulation and in the experiment?

=>=> The Cellular Potts parameters were chosen based on the average area and length of the cells. What are the values and how do they compare to experiments?

=>=> The amount of fibronectin was chosen so that there was a reasonable amount of bound VEGF. What is a reasonable amount? How did the authors compare to experiments?

=> line 399. The authors indicate that the results observed in figure 7 match the observations for elongated cells. And for the flattened cells? Why does the model do not reproduce their movement? Maybe this part needs more clarification by describing in greater detail how the comparison between the experimental and simulation results was carried out.

=> Figures 8 and 9. Here, the authors measure the number of junctions until it saturates, as a function of different parameters. However, it is nor clear for which number of junctions functional networks are obtained. If the number of junctions is large I presume that the cells are clustered together. The authors could analyse the different networks to identify the interval of number of junctions that correspond to networks, as it was done in Ramos et al, Phys Rev E 2018.

=> In figures 8 and 9 what are the size of the error bars? I.e. what variation can be expected in the number of junctions when running several simulations with the same parameters? Also, in figure 9 it is shown a prediction for the number of junctions for different values of VEGF concentration, c0. That experiment can be done in the lab. Does the simulation match the experimental results?

=> Besides the issues of the soluble VEGF binding in the apical region of the cell, I find the dynamics of bound VEGF problematic. In the simulation, bound VEGF concentration is larger underneath the cells. But how does it end up there? The diffusion of soluble VEGF through the matrix is hampered as the matrix is a porous media and also because of the VEGF binding sites. Therefore, by which mechanism does the soluble VEGF reach the underneath of the cell in such high concentration to bind? Moreover, there are three mechanisms that are not simulated that can alter this picture: secretion of VEGF by endothelial cells, intake of VEGF by endothelial cells, and production of MMPs that release VEGF from the matrix. Is it observed experimentally higher levels of bound VEGF underneath the cell? And if so, does that VEGF come from secretion of endothelial cells or from the soluble VEGF that finds more binding sites underneath the cell?

=> Line 544. Here, the authors mention that purely mechanical regulation is insufficient to model cell orientation and network formation. However, in van Oers et al 2014 the authors suggest a model that uses only mechanics to model similar mechanisms as in the present paper. From what I understand, here the simulated mechanism is the following: cell traction increases cell density under the cell, that leads to more binding of VEGF in the matrix, which drives the hapoptaxis and alignment of the cells which the soluble VEGF made elongated. In van Oers et al 2014 the forces elongate the cells and the cells migrate in the direction of larger strain, i.e. in the direction of other cells, with a similar effect of the VEGF hapt

---

## [Decision Letter · Decision Letter 1]

30 May 2024

Dear Prof Chung,

Thank you very much for submitting your manuscript "Simulation of Soluble and Bound VEGF-stimulated in vitro Capillary-like Network Formation on Deformed Substrate" for consideration at PLOS Computational Biology. As with all papers reviewed by the journal, your manuscript was reviewed by members of the editorial board and by several independent reviewers. The reviewers appreciated the attention to an important topic. Based on the reviews, we are likely to accept this manuscript for publication, providing that you modify the manuscript according to the review recommendations.

While of the reviewers were largely satisfied with the revisions and had only few remaining concerns that need to be addressed, one reviewer still finds the dynamics of bound VEGF highly problematic. As it has been shown that cells can accumulate ECM around them through pulling (Malandrino et al. 2019) I agree that it is not unreasonable to assume that bound VEGF could tag along with ECM and become concentrated as well. Nevertheless, the reviewer rightly comments that other processes might counteract such dynamics of VEGF, significantly affecting the model predictions and conclusions of your paper.  In response to their comments, you now write in the introduction that you do not consider such additional processes (l. 140-141). Additionally, in line with this reviewer’s comments, I would suggest that you address in the discussion section if, and in what way secretion and uptake of VEGF, as well as local diffusion and binding and unbinding of VEGF might affect your model predictions and how it could affect future work. 

Sincerely,

Roeland M.H. Merks, Ph.D

Academic Editor

PLOS Computational Biology

Jason Haugh

Section Editor

PLOS Computational Biology

Reviewer's Responses to Questions

**Comments to the Authors:**

Reviewer #1: Overall, the authors have adequately addressed my concerns. However, there are a couple of small things that still need to be fixed.

[1] In response to Rev 1 (E):

"Since cell culturing in incubators is at 37°C, our model does not include temperature as a variable. .. It will affect the cell movement and deformation, including shape and size."

And correction on L 583-584 of the paper.

This is a total misunderstanding by the authors of the CPM temperature parameter. This parameter has nothing to do with the lab temperature. It is just a quantity in CPM that controls the magnitude of fluctuations. It's OK to say "we kept the temperature parameter constant, T=1", but the lab temperature has nothing to do with it.

[2] More detailed figure captions were requested in the review. The revised paper makes it hard to find those captions, which are scattered between lines of text.

[3] The supplementary file is not in a very good shape. There are some weird notes like "MERGEFORMAT" right in the middle of the equations. Also, the equations are not fully readable. To be more specific:

 In equation (A2) it is hard to see the limits of the integral,

 In equation (A3) it is quite hard to read the subscripts,

 In equation (A8), in the Laplacian operator (nabla to the power 2): the power 2 is not very visible.

Reviewer #2: The authors have properly addressed my comments, and I would support publication.

Reviewer #3: In this work, the mechanism driving capillary network formation is the following: the cell traction increases matrix density under the cell, that leads to higher density of VEGF in the matrix under the cells, which drives the haptotaxis and alignment of the cells which have been made elongated. This mechanism depends on maintaining high VEGF levels bound to the matrix surface under the cells in in vitro assays.

The authors however did not explain satisfactorily in their reply how the VEGF is able to diffuse through the matrix and bind the ECM underneath the cells in high levels. I expect, otherwise, that the VEGF concentration in the matrix surface is higher in the regions where the cells are not located and are in direct contact with the soluble medium.

In their model, the authors do not account for the the space occupied by the cells and the VEGF diffuses through them. In this way, the model is not able to describe qualitatively the VEGF gradients at the cells' vicinity, which are essential to drive network formation, according to the authors' hypothesis.

Their work is thus based on these local surface-bound VEGF gradients that the model does not correctly model, and that the authors do not show experimental evidence they exist. For this reason I find that this work is not suitable for publication in PLoS Computational Biology.

**Have the authors made all data and (if applicable) computational code underlying the findings in their manuscript fully available?**

Reviewer #1: Yes

Reviewer #2: Yes

Reviewer #3: None

PLOS authors have the option to publish the peer review history of their article (what does this mean?). If published, this will include your full peer review and any attached files.

Reviewer #1: No

Reviewer #2: No

Reviewer #3: No

Figure Files:

Data Requirements:

Reproducibility:

References:

---

## [Editor Report · Decision Letter 2]

26 Jun 2024

Dear Prof Chung,

We are pleased to inform you that your manuscript 'Simulation of Soluble and Bound VEGF-stimulated in vitro Capillary-like Network Formation on Deformed Substrate' has been provisionally accepted for publication in PLOS Computational Biology.

Best regards,

Roeland M.H. Merks, Ph.D

Academic Editor

PLOS Computational Biology

Jason Haugh

Section Editor

PLOS Computational Biology

---

## [Editor Report · Acceptance letter]

5 Jul 2024

PCOMPBIOL-D-23-01988R2 

Simulation of Soluble and Bound VEGF-stimulated in vitro Capillary-like Network Formation on Deformed Substrate

Dear Dr Chung,

I am pleased to inform you that your manuscript has been formally accepted for publication in PLOS Computational Biology. Your manuscript is now with our production department and you will be notified of the publication date in due course.

With kind regards,

Olena Szabo
